# Seasonal Aerosol Acidity, Liquid Water Content and Their Impact on Fine Urban Aerosol in SE Canada

**Andrea M. Arangio** [1,*]**, Pourya Shahpoury** [2,3]**, Ewa Dabek-Zlotorzynska** [4] **and Athanasios Nenes** [1,5,*]

1    School of Architecture, Civil & Environmental Engineering, Ecole Polytechnique Fédérale de Lausanne, CH-1015 Lausanne, Switzerland

2    Air Quality Research Division, Environment and Climate Change Canada, 4905 Dufferin St., Toronto, ON M3H 5T4, Canada; pshahpoury@trentu.ca

3    Department of Chemistry, Trent University, 1600 West Bank Drive, Peterborough, ON K9L 0G2, Canada

4    Air Quality Research Division, Environment and Climate Change Canada, 335 River Road, Ottawa, ON K1A OH3, Canada; ewa.dabek@ec.gc.ca

5    Institute for Chemical Engineering Sciences, Foundation for Research and Technology Hellas, GR-26504 Patras, Greece

\*    Correspondence: andrea.arangio@epfl.ch (A.M.A.); athanasios.nenes@epfl.ch (A.N.)

**Abstract:** This study explores the drivers of aerosol pH and their impact on the inorganic fraction and mass of aerosol in the S.E. Canadian urban environments of Hamilton and Toronto, Ontario. We find that inter-seasonal pH variability is mostly driven by temperature changes, which cause variations of up to one pH unit. Wintertime acidity is reduced, compared to summertime values. Because of this, the response of aerosol to precursors fundamentally changes between seasons, with a strong sensitivity of aerosol mass to levels of $HNO_3$ in the wintertime. Liquid water content (LWC) fundamentally influences the aerosol sensitivity to $NH_3$ and $HNO_3$ levels. In the summertime, organic aerosol is mostly responsible for the LWC at Toronto, and ammonium sulfate for Hamilton; in the winter, LWC was mostly associated with ammonium nitrate at both sites. The combination of pH and LWC in the two sites also affects N dry deposition flux; $NO_3^-$ fluxes were comparable between the two sites, but $NH_3$ deposition flux at Toronto is almost twice what was seen in Hamilton; from November to March N deposition flux slows down leading to an accumulation of N as $NO_3^-$ in the particle phase and an increase in $PM_{2.5}$ levels. Given the higher aerosol pH in Toronto, aerosol masses at this site are more sensitive to the emission of $HNO_3$ precursors compared to Hamilton. For both sites, $NO_x$ emissions should be better regulated to improve air quality during winter; this is specifically important for the Toronto site as it is thermodynamically more sensitive to the emissions of $HNO_3$ precursors.

**Keywords:** atmospheric acidity; ammonium nitrate; $PM_{2.5}$; air quality

## 1. Introduction

Atmospheric acidity is a key parameter for particulate matter as it modulates, amongst other aspects, the gas–particle partitioning of semi-volatile ionizable species, and thus aerosol mass concentration. Aerosol pH was also shown to relate to the particulate matter health metrics [1,2]. Atmospheric aerosol acidity varies over five orders of magnitude in terms of $H^+$ molality (five units of pH) due to variations in both meteorological variables (temperature and relative humidity) and aerosol chemical compositions. Tao et al. [3] showed that in six Canadian cities over a period of 10 years, changes in ambient temperature largely drive the seasonality of aerosol pH. The study indicates that while during summertime, aerosol pH (1–2) is acidic with small variation, pH during wintertime is less acidic (3–4) with larger variation, particularly in sites with lower $NH_4^+:SO_4^{2-}$ molar ratios. Lawal et al. [4] analyzed the time series of aerosol pH in six regions within the United States and showed that pH variations are not significant despite the emissions

reduction in important aerosol precursors such as sulfur dioxide and nitrogen oxide. Upon oxidation, sulfur and nitrogen oxides lead to the formation of gaseous species such as sulfuric acid ($H_2SO_4$) and nitric acid ($HNO_3$) that, together with organic acids, tend to increase aerosol acidity. Species such as ammonia ($NH_3$) and non-volatile cations (NVCs: $Na^+$, $K^+$, $Ca^{2+}$, $Mg^{2+}$) decrease aerosol acidity. Together with aerosol liquid water content (LWC), aerosol acidity (pH) governs the gas–particle partitioning of semi-volatile gases and bases such as $NH_3$ and $HNO_3$ [5–9]. Because of its semi-volatile nature, gas-phase nitric acid partitions to the particle phase only in moderately acidic to neutral conditions, eventually reacting with $NH_3$ to form ammonium nitrate or with inorganic cations found in sea salt, mineral dust, and biomass burning to form a variety of inorganic (soluble) salts. In acidic environments, gas-phase $NH_3$ reacts promptly with $H_2SO_4$ and $HNO_3$ to form particle phase ammonium sulfate/bisulfate and nitrate [10]. In both scenarios, ammonium and nitrate salts constitute an important fraction of ambient $PM_{2.5}$ mass [11–13]. Given that, for dry deposition, species in the gas phase generally have a shorter atmospheric residence time than those in the particulate phase [10], the degree of gas–particle partitioning can directly impact the atmospheric residence time of nitrogen ("reactive nitrogen", Nr) species, with important implications for particulate matter levels in the boundary layer and dry deposition [14,15]. It is therefore important to consider how aerosol acidity and liquid water content, through their effect on gas–particle partitioning, affect the deposition fluxes of Nr species. In this study, we applied the thermodynamic framework developed by Nenes et al. [9,16] to characterize the links between deposition of Nr, aerosol acidity, and accumulation of $NO_3^-$ in the boundary layer over two Canadian cities, Toronto and Hamilton, by identifying the relevant "chemical regimes" and their effects on deposition flux and levels of PM.

## 2. Methods

### 2.1. Study Location

PM$_{2.5}$ samples were collected from the National Air Pollution Surveillance (NAPS) monitoring sites in Toronto and Hamilton, which are classified as large urban areas according to the NAPS classification framework [17]. The sampling site (NAPS site ID: 060438) in Toronto is classified as near-road, being strongly influenced by transportation sources, while Hamilton is influenced by a point source (steel factory). Land use classifies the Toronto site as commercial, and Hamilton as residential. With respect to population, the site in Toronto is classified as highly populated ($\geq$150,000), whereas Hamilton is in the mid-population range (100,000–149,999).

### 2.2. PM$_{2.5}$ Sample Collection and Analysis

A total number of 251 samples were collected from Hamilton between 2016 and 2018, and, 344 samples were collected in Toronto up to 2019. The alkaline ($NH_3$) and acidic ($HNO_3$, $SO_2$) gaseous components were collected using a citric acid-coated and $Na_2CO_3$-coated honeycomb glass denuders, respectively. The series of two denuders was followed by two cassettes containing PTFE and Nylon filters to collect PM$_{2.5}$ and volatile nitrate accordingly [18–20]. At each site, 24 h samples were collected once every three days, resulting in the sampled air volumes of 14.4 m$^3$. The denuder coating and PM$_{2.5}$ were extracted in water and analyzed for water-soluble anions and cations using ion chromatography (IC, Thermo Scientific, Sunnyvale, CA, USA). Field blanks were routinely collected and used for background corrections. Dabek-Zlotorzynska et al. (2011) [18] provide the detailed protocol for ambient constituent measurements for the NAPS database.

### 2.3. Estimation of Aerosol pH

Aerosol pH is obtained through "thermodynamic analysis" of the ambient observations [1,7,21], where a thermodynamic model is applied to observations of the major inorganic species in the gas and particulate phase, to determine the equilibrium liquid water content and hydrogen ion concentration required for calculation of pH. The model is

said to provide realistic estimates of acidity if the predicted partitioning of semi-volatile acid/basic species (e.g., NH$_3$, HCl, and HNO$_3$) and liquid water content (LWC) matches the observed values. Measurements of LWC are seldomly available [7,22] (including this study), but models tend to adequately predict the quantity for RH above 40% [8], where the assumption of metastable aerosol often also applies. Therefore, we consider only data for which RH exceeds 30% and evaluate only the semi-volatile partitioning to ensure that pH is reasonable.

In this study, we applied the ISORROPIA-lite (http://isorropia.epfl.ch, n.d., (accessed on 15 May 2022)) model [23,24]; to obtain the equilibrium concentration of H$^+$ and LWC in the aerosol, and calculated the pH using the "pH$_F$" definition of Pye et al. (2020) [5],

$$\mathrm{pH} = -\log_{10}\gamma_{\mathrm{H^+}} \cong -\log_{10}\frac{1000\gamma_{\mathrm{H^+}}\mathrm{H_{air}^+}}{\mathrm{W_i + W_o}}$$

where $\gamma_{\mathrm{H+}}$ is activity coefficient of the hydronium ion, H$^+$ (assumed unity), H$_{aq}{}^+$ is its concentration (mol L$^{-1}$) in the aerosol aqueous phase, H$_{air}{}^+$ ($\mu$g m$^{-3}$) is the concentration of H$^+$ per volume of air, and W$_i$, W$_o$ ($\mu$g m$^{-3}$) is particle water concentrations associated with the aerosol inorganic, and organic species, respectively. The organic fraction perturbs the partitioning of the semi-volatiles (such as nitrate, ammonium, and chloride) and aerosol pH moderately (between 0.15 and 0.30 units [7,25–27]). ISORROPIA-lite considers the effect of the organic fraction by specifying the hygroscopic parameter kappa (0.15), the density of the PM organic fraction (1.0 mg/mL), and by calculating the contribution to LWC, which ultimately affects the aerosol pH.

## 3. Results and Discussion

### 3.1. Seasonality of Aerosol Acidity

Figure 1 shows the time series of daily aerosol pH (black dots) and monthly average pH for both Hamilton (red) and Toronto (light blue) and monthly standard deviation for the whole observational periods (Hamilton: from 1 January 2016 to 31 December 2017; Toronto: 1 January 2016 to 31 December 2018).

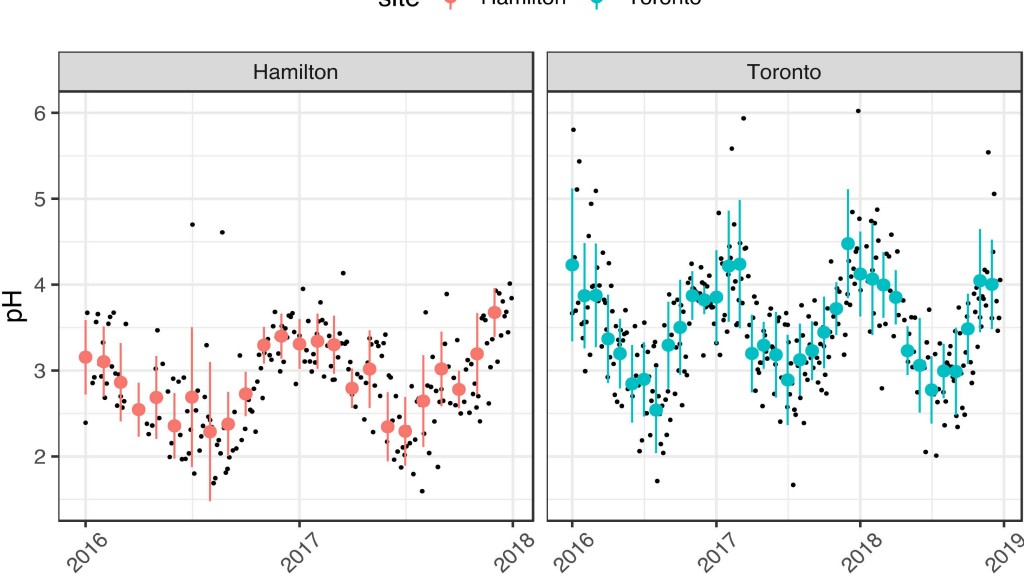

**Figure 1.** Aerosol pH (black dots) and monthly average pH in Hamilton (red) and Toronto (light blue) over the period of 2016 to 2019. Error bars are standard deviation of monthly pH distribution at the two sites.

The reliability of aerosol pH calculation is evaluated by comparing the modeled and the measured gas–particle partitioning of $NH_3$. Figure S1 shows that the modeled partitioning of $NH_3$ reproduces consistently the observations ($R^2 = 0.97$), indicating that the pH estimates shown in Figure 1 are robust.

In both sites, aerosol acidity is characterized by a strong seasonality. The lowest aerosol pH recorded in summer is comparable between the two sites on both daily basis (1.6 in Hamilton and 1.7 in Toronto recorded in July 2017) and monthly basis (mean ± standard deviation (SD): 2.3 ± 0.8 in Hamilton and 2.5 ± 0.5 in Toronto recorded in August 2016). The highest aerosol pH values are recorded in winter and they differ between the sites with larger values seen in Toronto (daily pH = 6 and monthly pH = 4.5 ± 0.6 in December 2017); these are nearly 1 pH unit higher than those seen in Hamilton (daily pH = 4.7 and monthly pH = 3.6 ± 0.3 in December 2017). The higher aerosol winter pH in Toronto results in a larger seasonal variability compared to Hamilton, i.e., a difference of ~2 pH units compared to ~1.3 units, respectively (considering the monthly averages). Tao et al. [3] have reported similar seasonality of aerosol pH in Toronto, which has been attributed to annual variations in both meteorological parameters and PM chemical composition. In their work, pH values in Toronto ranged between 1.7 in summer and 3.6 during winter, with larger variability seen during winter. While the pH range for Hamilton in the present work is comparable to that reported by Tao et al., the range calculated here for Toronto is ~1 pH unit higher. This difference may be attributed to the additional contribution by the aerosol organic fraction that was taken into account in this work; this leads to an increase in aerosol pH, as described in the following sessions.

To evaluate the sensitivity of seasonal pH to temperature, the aerosol pH was computed by setting the temperature equal to the overall dataset average temperature (11 and 10 °C for Hamilton and Toronto, respectively). The resulting pH is then compared to the aerosol pH computed considering the actual temperature. The difference between the two resulting values, ΔpH, is shown in Figure 2 on a monthly base.

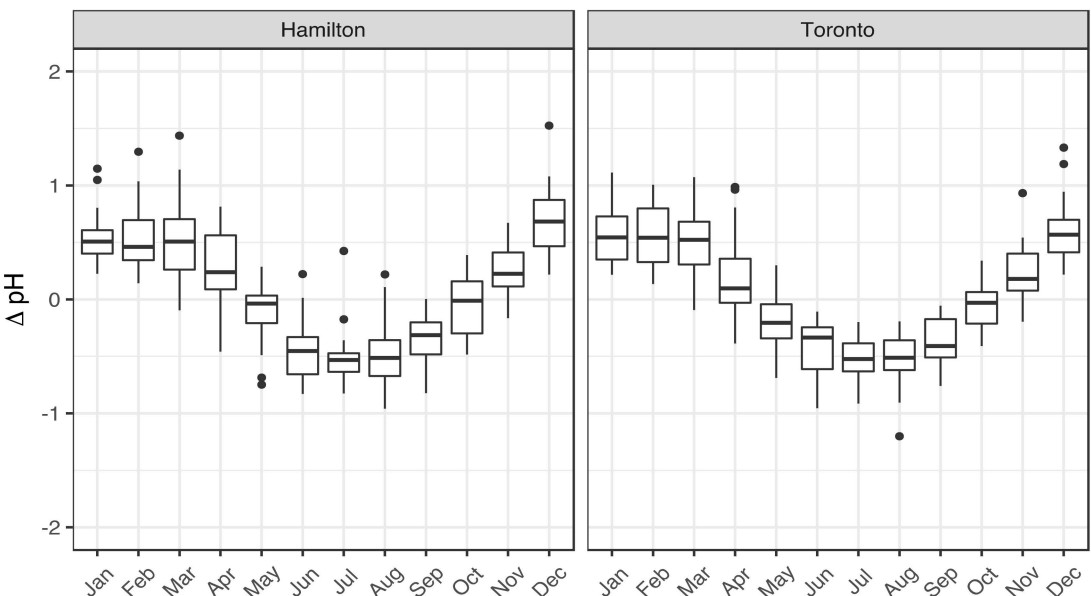

**Figure 2.** Sensitivity of pH to temperature—expressed as difference between the pH calculated using ambient values of RH and T and the pH calculated assuming a constant, average annual temperature of 11 °C for Hamilton and 10 °C for Toronto.

In both sites, the median ΔpH values are up to about +0.5 units in Winter and up to −0.5 units in Summer. The result indicates that temperature variation accounts for up to 1 unit variation in aerosol pH, increasing its value in winter and decreasing it in summer. Indeed, low temperatures promote the partitioning of semi-volatile compounds

such as nitric acid, ammonia, and water to the particulate phase. The extra water-soluble material further equilibrates with water vapor leading to additional water condensation and decreasing the $H^+$ concentration [28]; the opposite is observed in summer.

The seasonal variability in acidity ranges up to 1 pH unit, and is somewhat larger in winter and smaller in summer in Toronto site, and is the opposite in Hamilton; this can be attributed to variations in aerosol chemical composition [3]. The higher pH levels (and variability) in Toronto are driven by the combined effect of low temperatures in winter and a large fraction of traffic-related organic aerosol (see Table 1), which increase the aerosol LWC.

**Table 1.** Chemical composition of major aerosol particles, gas-phase species, and aerosol pH in Hamilton and Toronto. For all chemical species, seasonal average and standard error of concentrations are reported in $\mu g\ m^{-3}$. The significance (*p*-value) of the difference in seasonal concentration of each variable in the two sites is calculated using a two-way ANOVA test with a significance level of 0.05.

| Variable | Season | Hamilton | Toronto | Significance (*p*-Value) |
|---|---|---|---|---|
| Gas–$NH_3$ ($\mu g\ m^{-3}$) | Winter | $1.17 \pm 0.02$ | $3.17 \pm 0.02$ | <0.05 |
| | Spring | $2.66 \pm 0.05$ | $3.93 \pm 0.02$ | <0.05 |
| | Summer | $3.27 \pm 0.02$ | $4.66 \pm 0.02$ | <0.05 |
| | Fall | $3.53 \pm 0.09$ | $4.68 \pm 0.03$ | <0.05 |
| Particle–$NH_4^+$ ($\mu g\ m^{-3}$) | Winter | $0.94 \pm 0.01$ | $1.02 \pm 0.01$ | 0.50 |
| | Spring | $0.71 \pm 0.01$ | $0.47 \pm 0.01$ | <0.05 |
| | Summer | $0.42 \pm 0.01$ | $0.32 \pm 0.01$ | 0.14 |
| | Fall | $0.42 \pm 0.01$ | $0.37 \pm 0.01$ | 0.38 |
| Total–$NH_4$ ($\mu g\ m^{-3}$) | Winter | $2.10 \pm 0.02$ | $4.19 \pm 0.03$ | <0.05 |
| | Spring | $3.37 \pm 0.06$ | $4.40 \pm 0.03$ | <0.05 |
| | Summer | $3.69 \pm 0.02$ | $4.99 \pm 0.02$ | <0.05 |
| | Fall | $3.96 \pm 0.09$ | $5.05 \pm 0.03$ | <0.05 |
| Gas–$HNO_3$ ($\mu g\ m^{-3}$) | Winter | $0.10 \pm 0.01$ | $0.03 \pm 0.001$ | <0.05 |
| | Spring | $0.24 \pm 0.005$ | $0.18 \pm 0.01$ | 0.14 |
| | Summer | $0.46 \pm 0.003$ | $0.49 \pm 0.01$ | 0.49 |
| | Fall | $0.26 \pm 0.003$ | $0.14 \pm 0.002$ | <0.05 |
| Particle–$NO_3^-$ ($\mu g\ m^{-3}$) | Winter | $2.29 \pm 0.03$ | $2.85 \pm 0.03$ | 0.15 |
| | Spring | $0.92 \pm 0.02$ | $0.96 \pm 0.02$ | 0.14 |
| | Summer | $0.09 \pm 0.02$ | $0.20 \pm 0.02$ | <0.05 |
| | Fall | $0.54 \pm 0.02$ | $0.64 \pm 0.01$ | 0.51 |
| Total–$NO_3$ ($\mu g\ m^{-3}$) | Winter | $2.39 \pm 0.03$ | $2.87 \pm 0.03$ | 0.21 |
| | Spring | $1.17 \pm 0.02$ | $1.14 \pm 0.02$ | 0.89 |
| | Summer | $0.55 \pm 0.003$ | $0.70 \pm 0.004$ | <0.05 |
| | Fall | $0.81 \pm 0.01$ | $0.78 \pm 0.01$ | 0.85 |
| $SO_4^{2-}$ ($\mu g\ m^{-3}$) | Winter | $1.18 \pm 0.01$ | $1.09 \pm 0.01$ | 0.33 |
| | Spring | $1.71 \pm 0.03$ | $0.98 \pm 0.01$ | <0.05 |
| | Summer | $1.51 \pm 0.02$ | $1.12 \pm 0.02$ | 0.05 |
| | Fall | $1.15 \pm 0.02$ | $0.90 \pm 0.01$ | 0.12 |
| OM ($\mu g\ m^{-3}$) | Winter | $1.68 \pm 1.88$ | $3.92 \pm 0.67$ | <0.05 |
| | Spring | $1.68 \pm 1.97$ | $4.56 \pm 0.74$ | <0.05 |
| | Summer | $2.48 \pm 2.37$ | $6.91 \pm 0.51$ | <0.05 |
| | Fall | $1.72 \pm 2.62$ | $5.43 \pm 0.61$ | <0.05 |
| pH | Winter | $3.34 \pm 0.37$ | $4.08 \pm 0.62$ | <0.05 |
| | Spring | $2.92 \pm 0.45$ | $3.57 \pm 0.59$ | <0.05 |
| | Summer | $2.44 \pm 0.60$ | $2.91 \pm 0.47$ | <0.05 |
| | Fall | $2.90 \pm 0.45$ | $3.50 \pm 0.53$ | <0.05 |

Table 1 compares the average concentrations of main species that contribute to aerosol acidity across various seasons. The most significant differences between the two sites are

the concentration of gas-phase $NH_3$ and the amount of organic matter in the particulate phase. According to the NAPS site classification framework, and source apportionment in our previous study [2], Toronto is classified as a transportation-influenced site [17] being located close to a major roadway, while Hamilton is influenced by industrial activities (steel production). Therefore, traffic-related emissions explain the high $NH_3$ concentration and OM fraction in Toronto compared to Hamilton. Gasoline and diesel vehicles equipped with catalyst or selective catalytic reduction emit large amounts of $NH_3$ [29]. The difference in gas phase $NH_3$ concentrations between the two sites is statistically significant during winter, spring, and summer, with the winter concentration in Toronto reaching more than twice the concentration in Hamilton during the same season. The differences in the amounts and seasonal variation in the total reduced nitrogen (total $NH_4$–gas phase $NH_3$ and particle phase $NH_4^+$) over the two sites confirm the local nature of this pollutant in Toronto being continuously emitted throughout the year. Indeed, while in Hamilton total $NH_4$ increases by more than 45% from its minimum value in winter (2.10 µg m$^{-3}$) to its maximum in fall (3.96 µg m$^{-3}$), in Toronto, the total $NH_4$ increases by ~15% (from 4.19 to 5.05 µg m$^{-3}$). This small variation throughout the year might be caused by constant emissions from a source such as road traffic. On the other hand, at both sites, the total oxidized nitrogen (total $NO_3$–gas phase $HNO_3$ and particle phase $NO_3^-$) changed by ~75% from its minimum value in summer to its maximum value in winter; this indicates a regional character of the pollutant. Vehicular emissions provide precursors for the formation of secondary organic aerosol (SOA), which contributes to OM. The significant difference in OM concentrations between the two sites (with larger OM fractions in Toronto throughout the year) highlights the importance of traffic-related emissions at the Toronto site. Combustion of fossil fuels and smelters are primary sources of $SO_2$ [30–32], which is rapidly oxidized by •OH in the gas phase, or hydrogen peroxide in the aerosol aqueous phase to form particulate $SO_4^{2-}$ [10]. The concentrations of particulate $SO_4^-$ are not statistically different at the two sites, indicating the regional character of this pollutant.

Particle liquid water content (LWC) is a key parameter that determines the partitioning of semi-volatile inorganic species and pH (e.g., Nenes et al., 2020 [9]) and depends on the RH, temperature, and the equilibrium aerosol composition. Some chemical species in the aerosols are more hygroscopic than others, such as chloride salts, but the main driver of LWC is often $SO_4^{2-}$ as it is in larger amounts compared to the other inorganic aerosol species. Large amounts of $SO_4^{2-}$ contribute also to increase the $H^+$ concentration leading to strongly acidic aerosol [5,8]. Other atmospherically relevant ions such as $NH_4^+$ and $NO_3^-$, nonvolatile cations (NVC: $Na^+$, $K^+$, $Mg^{+2}$, and $Ca^{+2}$), and OM also contribute to particle LWC, without increasing the acidity considerably; therefore, they tend to elevate the aerosol pH [8]. In Figure 3, the monthly average contribution of particulate inorganic salts and organic fraction to the LWC, as predicted by ISORROPIA-lite, are reported for Hamilton (left) and Toronto (right) and expressed as percentage of water speciated for chemical species. LWC in Hamilton is largely dominated by the inorganic fraction throughout the year with $NH_4NO_3$ contributing up to 40% from November to February and dropping down to less than 10% during the rest of the year and $(NH_4)_2SO_4$ contributing to the 25% from November to February and up to 55% from March to October. The contribution of the PM organic fraction to LWC remains below 30% throughout the year. In Toronto, the organic fraction contributes up to 65% to the LWC with the highest values seen between May and November; this is followed by $(NH_4)_2SO_4$, which contributed 20–30% to LWC in the same period. Although organics remain an important contributor taking up between 20 and 35% of the water from December to March, the major contributor to particle LWC in this period is $NH_4NO_3$ (~40%). Other inorganic salts such as $(NH_4)_2SO_4$, $NH_4Cl$, $Na_2SO_4$, and NaCl contribute to about 12, 8, 6, and 2% of water, respectively, in the same period. The plots in Figure 3 report also the absolute amount of water in red circles on the right *y*-axis. In both sites, aerosol LWC is high from December to March and low during the rest of the year with Toronto presenting a stronger seasonality compared to Hamilton. During the winter months, the temperature is on average about 0 °C and rises up to an average of

16 °C during the remaining months. Together with water vapor condensation, low winter temperatures favor the partitioning of semi-volatile inorganic species such as $NH_3$ and $HNO_3$, which in turn contributes to increasing the LWC. As a result of these two combined effects, particle LWC reaches its maximum during winter months and its minimum during summer when the non-volatile organic fractions are the main aerosol component.

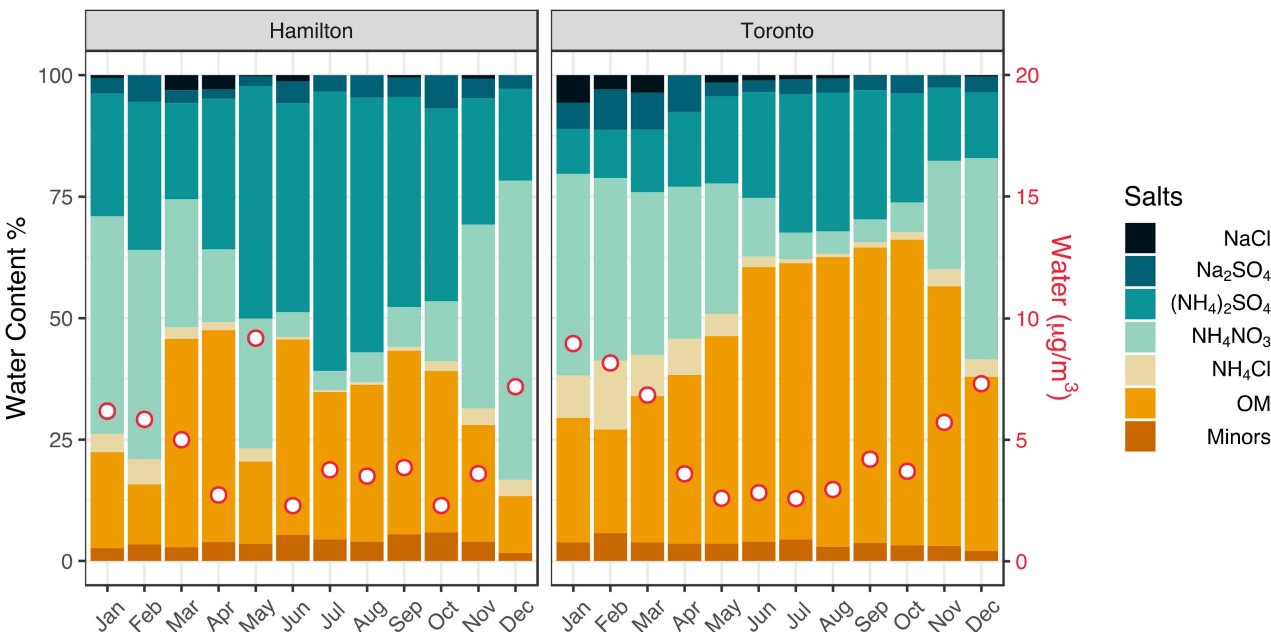

**Figure 3.** Monthly average of aerosol Liquid water content (LWC) in Hamilton (**left**) and Toronto (**right**) expressed as a percentage of water content speciation per chemical salt contained in the particulate phase (right *y*-axis); absolute aerosol LWC express in $\mu g\ m^{-3}$.

Together with $SO_2$, gas-phase ammonia and nitric acid are amongst the most important aerosol precursors in terms of aerosol mass upon conversion to their non-volatile salts [12,13]. $NH_3$ is the most important alkaline species in the atmosphere that contributes to particulate matter[10] while nitrate originates mostly from the oxidation of $NO_x$ emitted from combustion sources. In un-dissociated form, $NH_3$ and $HNO_3$, are highly volatile, while they are nearly non-volatile in their ionic forms ($NO_3^-$, $NH_4^+$). The prominence of species in their volatile or nonvolatile forms is driven by acid-base equilibria with the particulate matter. Specifically, in strongly acidic aerosol (typically with pH <1.5 to 2), nitrate remains mostly in the gas-phase as $HNO_3$ (e.g., Nenes et al., 2020 [9]) and vice-versa for mildly acidic conditions (typically pH > 3). Compared to nitrate, ammonia exhibits an opposite behavior with respect to aerosol pH. For these reasons, aerosol pH and liquid water content (LWC) are key parameters that control $NH_3$ and $HNO_3$ gas–particle partitioning and ultimately the particulate matter sensitivity to their total concentration [9]. Meskhidze et al. (2003) [6] and later Guo et al. (2017) [8] showed the relation that explicitly links the gas–particle partitioning for both $NO_3^-$ and $NH_4^+$, respectively $\varepsilon(NO_3^-)$ and $\varepsilon(NH_4^+)$, with particle phase concentration of $H^+$, $[H^+]$, particle LWC ($W_i$), and air temperature (T):

$$\varepsilon\left(NO_3^-\right) = \frac{K_{n1}\,H_{HNO_3}\,W_i\,RT}{\gamma_{H^+}\gamma_{NO_3^-}\left[H^+\right] + K_{n1}\,H_{HNO_3}\,W_i\,RT} \quad \text{and} \quad \varepsilon\left(NH_4^+\right) = \frac{\frac{\gamma_{H^+}}{\gamma_{NH_4^+}}\frac{H_{NH_3}}{K_a}\left[H^+\right]W_i\,RT}{1 + \frac{\gamma_{H^+}}{\gamma_{NH_4^+}}\frac{H_{NH_3}}{K_a}\left[H^+\right]W_i\,RT} \tag{1}$$

where $H_{HNO_3}$ and $H_{NH_3}$ and Henry's law constants for $HNO_3$ and $NH_3$, respectively; $K_{n1}$ and $K_a$ are the acid dissociation constants for $HNO_3$ and $NH_4^+$; R is the universal gas

constant; and $\gamma_{H^+}$, $\gamma_{NO_3^-}$ and $\gamma_{NH_4^+}$ are the single-ion activity coefficients for $H^+$ and $NO_3^-$, respectively.

In their framework, Nenes et al. [9] applied equation 1 to determine a "characteristic pH" for $NH_3$ and $HNO_3$ as a function of air temperature, relative humidity, pH, and LWC, which is associated with a shift in sensitivity of aerosol mass to changes in $NH_3$ and $HNO_3$. The characteristic pH is described as $pH' = -\log\left[\frac{1-\alpha}{\alpha}\Psi W_i T\right]$ for $HNO_3$ and $pH'' = -\log\left[\frac{1-\beta}{\beta}\Phi W_i T\right]$ for $NH_3$, where $\Psi = RTK_{n1}H_{HNO_3}/\gamma_{H^+}\gamma_{NO_3^-}$ and $\Phi = \frac{\gamma_{H^+}}{\gamma_{NH_4^+}}\frac{H_{NH_4^+}}{K_a}RT$. The parameters $\alpha$ and $\beta$ are arbitrary threshold factors set equal to 0.1 under the assumption that aerosol responds to changes in $NH_3/HNO_3$ when at least 10% of the gas-phase precursors is partitioned to the particle phase. Figure 4 shows the aerosol sensitivity map resulting from the application of the framework to the Hamilton and Toronto datasets. The "characteristic pH" for $NH_3$ (blue diagonal line) and $HNO_3$ (red diagonal line) are calculated as a function of aerosol LWC, considering an average temperature (11 °C) and relative humidity (68%) from the combination of the two datasets. In Figure 4, daily aerosol pH is reported against the corresponding LWC value for Hamilton (red) and Toronto (light blue). In both sites, all points lay almost entirely above the red line, indicating that these combinations of aerosol pH and LWC favor further $HNO_3$ molecules to be partitioned in the particle phase as $NO_3^-$. Points laying also above the blue line indicate conditions of pH and LWC that favor $NH_3$ to the gas phase.

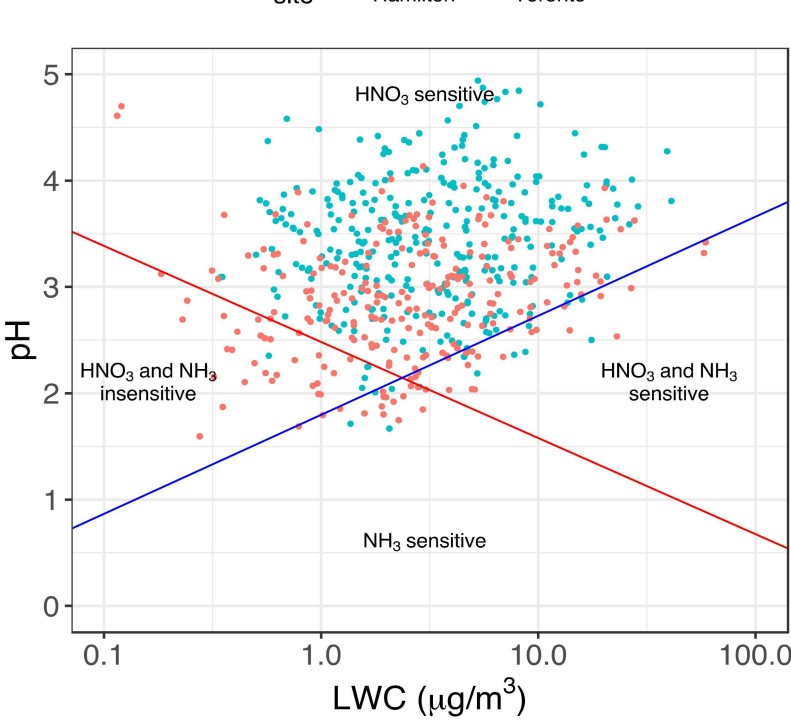

**Figure 4.** Domains of aerosol sensitivity to ammonia and nitrate levels calculated for the average temperature of the respective dataset. The values for Hamilton are shown in red, and those for Toronto are shown in light blue. The red and blue diagonal lines indicate the "characteristic pH" for $HNO_3$ and $NH_3$, respectively, estimated using the average temperature (11 °C) and RH (68%) for the two combined datasets.

The sensitivity conditions can be estimated more precisely by calculating the characteristic pH on daily basis and by subtracting the actual aerosol pH from the characteristic pH. The resulting $\Delta$pH is a measure of how far the actual pH is from the threshold value of the characteristic pH. The time series of $\Delta$pH for $HNO_3$ and $NH_3$ are reported on the y-axes

in Figure 5a,b, respectively. The $\Delta$pH for $HNO_3$ has a strong seasonal cycle, with higher values in winter for both sites indicating high sensitivity of PM to $HNO_3$ concentration during this period. In the summertime, the $\Delta$pH decreases to reach negative values (points below the solid black line in Figure 5a) and aerosol masses become insensitive towards $HNO_3$. Under such conditions, aerosol acidity and LWC favor $HNO_3$ volatilization. On the other hand, $\Delta$pH for $NH_3$ does not exhibit the seasonal trend at both sites. Aerosol pH and LWC favor $NH_3$ to be partitioned mostly in the gas-phase; consequently, aerosol mass is not sensitive to a further increase in $NH_3$ emissions.

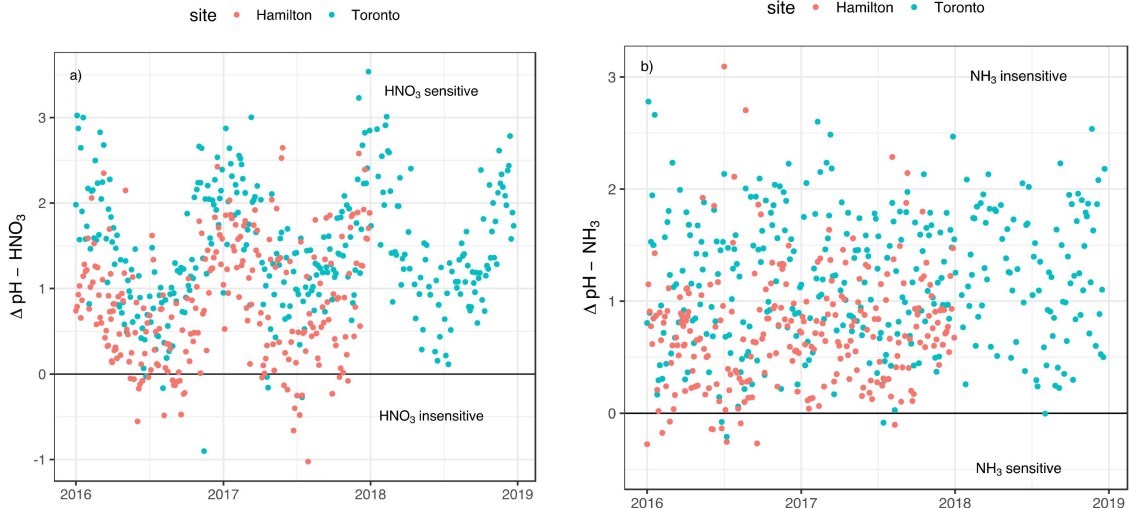

**Figure 5.** Time series of calculated $\Delta$pH for (**a**) $HNO_3$, and (**b**) $NH_3$ in Hamilton (red) and Toronto (light blue).

Based on $\Delta$pH reported in Figure 5a,b, the particulate matter sensitivity to the gas phase $HNO_3$ and $NH_3$ is different at the two sites over the same period. The aerosol is insensitive to both compounds for ~10% of the sampling days in Hamilton, whereas this value drops to 2% in Toronto given the higher pH at this site.

### 3.2. Implication of Aerosol Acidity Variability on Nitrogen Dry Deposition

Gas-phase $NH_3$ and $HNO_3$ tend to have a shorter residence time in ambient air compared to particle phase $NH_4^+$ and $NO_3^-$ [10]. Aerosol acidity and particle LWC modulate the gas–particle partitioning of $NH_3$ and $HNO_3$ and, as a result, their atmospheric residence time and amounts that can accumulate in the boundary layer [16]. The thermodynamic framework developed by Nenes et al. [16] describes how pH and LWC govern the dry deposition pattern of reduced and oxidized nitrogen species; this gives rise to "rapid" or "slow" deposition for each species, and consequently defines four possible domains. The total estimated deposition flux for $HNO_3$ (FNO3T) and $NH_3$ (FNH3T) is defined as the sum of their gas and particle concentrations at the equilibrium multiplied respectively by their gas and particle deposition velocity. Specifically, FNO3T = vg CHNO3 + vp CNO3$^-$ and FNH3T = vg CNH3 + vp CNH4+ where $v_g$, $v_p$ are the gas and particle deposition velocities, respectively; CHNO3 and CNH3 are the gaseous concentrations and CNO3$^-$ and CNH4+ are the particulate concentrations of $NH_3$ and $HNO_3$, respectively. If $NO_3^T$ and $NH_3^T$ represent the total concentrations of $NH_3$ and $HNO_3$, respectively, in the air mass, $k$ is the ratio between gas and particle deposition velocities, $FNO_3^T$ and $FNH_3^T$ is given by:

$$F_{NO_3^T} = v_P\left[k + (1-k)\varepsilon\left(NO_3^-\right)\right]NO_3^T \text{ and } F_{NH_3^T} = v_P\left[k + (1-k)\varepsilon\left(NH_4^+\right)\right]NH_3^T \quad (2)$$

The estimated nitrogen fluxes of $NH_3$ and $HNO_3$ at the two sites are reported as seasonal averages and compared statistically in Table 2. In Hamilton, the estimated fluxes of the

two species are comparable on average over the whole year (FNO3T = 29.2 µmol m$^{-2}$ s$^{-1}$; FNH3T = 32.5 µmol m$^{-2}$ s$^{-1}$). In Toronto, the estimated yearly averaged deposition flux of ammonia is about twice (FNH3T = 55.5 µmol m$^{-2}$ s$^{-1}$) the deposition flux of nitrate (FNO3T = 23.2 µmol m$^{-2}$ s$^{-1}$). In both sites, NO$_3^-$ tends to accumulate in the particle phase, leading to comparable deposition fluxes. Because of the higher pH and NH$_3$ abundance in the gas-phase, the estimated deposition fluxes of NH$_3$ in Toronto are two times larger than those in Hamilton. Nonetheless, at both sites, FNH3T increases from winter to fall with maximum value during summer, consistent with the gas-phase NH$_3$ concentrations, while NH$_4^+$ particle concentration has an opposite trend, and reaches its minimum during summer due to gas-particle equilibration driven by temperature variations. The sum of NH$_3$ and HNO$_3$ deposition fluxes is larger in Toronto site than in Hamilton.

**Table 2.** Seasonal estimates of N deposition fluxes for HNO$_3$ and NH$_3$ calculated for Toronto and Hamilton. The seasonal averages are compared using the ANOVA test to evaluate the significance of their statistical differences. The resulting *p*-values less than 0.05 indicate a statistically significant difference between the two values.

| | FNO3T (µmol m$^{-2}$ s$^{-1}$) | | | FNH3T (µmol m$^{-2}$ s$^{-1}$) | | |
|---|---|---|---|---|---|---|
| | **Toronto** | **Hamilton** | ***p*-Value** | **Toronto** | **Hamilton** | ***p*-Value** |
| Winter | 30.1 | 40.3 | <0.05 | 50.0 | 23.9 | <0.05 |
| Spring | 22.6 | 24.5 | 0.357 | 54 | 25.8 | <0.05 |
| Summer | 26.2 | 26.5 | 0.889 | 60.6 | 42.4 | <0.05 |
| Fall | 14.0 | 25.4 | <0.05 | 57.5 | 38.1 | <0.05 |
| Average | 23.2 | 29.2 | | 55.5 | 32.5 | |

In addition, considering F$_{N_r}$ as the sum of *FNO3T* and *FNH3T*, the non-dimensional flux for reactive nitrogen (N$_r$) can be derived as follow:

$$F_{N_r}^* = \frac{F_{N_r}}{v_p N_r} = (1-k)\left\{\varepsilon\left(NH_4^+\right) - \varepsilon\left(NO_3^-\right)\right\}\Gamma + \left\{k + (1-k)\varepsilon\left(NO_3^-\right)\right\} \tag{3}$$

where Nr is the total reactive nitrogen concentration, $\Gamma = \frac{NH_3^T}{N_r}$ is the fraction of Nr in the form of NH$_3^T$. F$_{N_r}^*$ indicates how fast is the actual estimated deposition compared to the case where the deposition is entirely determined by estimated particle deposition and therefore has a direct impact on the atmospheric residence time of aerosol and its precursors. F$_{N_r}^*$ ranges between 1 and 10 in the assumption that the maximum difference between gas and particle deposition velocities is equal to 10 [10,33]. Values of F$_{N_r}^*$ close to one indicate that the deposition occurs mostly by particle deposition, the atmospheric residence time of the species increases, and it accumulates in the boundary layer. Inversely, F$_{N_r}^*$ values close to 10 indicate that the fast gas phase deposition dominates the removal of the precursors, reducing the atmospheric residence time of the species in the boundary layer.

Figure 6 shows the monthly average values of F$_{N_r}^*$ for Hamilton (red) and Toronto (light blue). For both sites, the median values of F$_{N_r}^*$ (black horizontal line in each vertical bar) span between 6 and 8 during the winter months and it approaches 10 from April to November.

From December to March, the partitioning of N to the particle phase increases up to about three times compared to summer months due to the variation in pH and LWC throughout the year and leads to the accumulation on N in the boundary layer. The monthly averages of NO$_3^-$ in the particle phase and HNO$_3$ in the gas phase in the two sites, reported in b and 6d, change according to F$_{N_r}^*$. Indeed, the concentration of NO$_3^-$ in the particle phase is low in summer and reaches its maximum within December and March; this is when the gas phase HNO$_3$ is at its minimum (HNO$_3$ peaks during summer months). Particle phase NO$_3^-$ concentration is determined by the combination of the total NO$_3^-$ concentration in the air masses, which primarily depends on NO$_3^-$ production

(photolysis) and loss rate (deposition velocity), and the gas-particle partitioning coefficient $\varepsilon_{NO3}$ (ranging from 0 to 1), which is the function of T, pH, and LWC. $\varepsilon_{NO3}$, reported in c for the two sites, remains larger than 0.5 during the whole winter with values that are larger for Toronto than for Hamilton. The slow pH-driven deposition flux in winter contributes directly to the accumulation of N in the boundary layer as $NO_3^-$. The opposite patterns in particle and gas-phase concentrations of $NO_3^-$ and $HNO_3$ at the two sites during this period are determined by the differences in aerosol pH. The less acidic aerosol in Toronto leads to an increased partitioning of $NO_3^-$ to the particulate phase, as shown by the increase in $\varepsilon_{NO3}$. In these thermodynamic conditions, hypothetical increment of $NO_3^-$ precursors and $NO_3^-$ production lead to a further increase in its partitioning to the particulate phase, which will result in condensing more water and increasing the pH and aerosol mass.

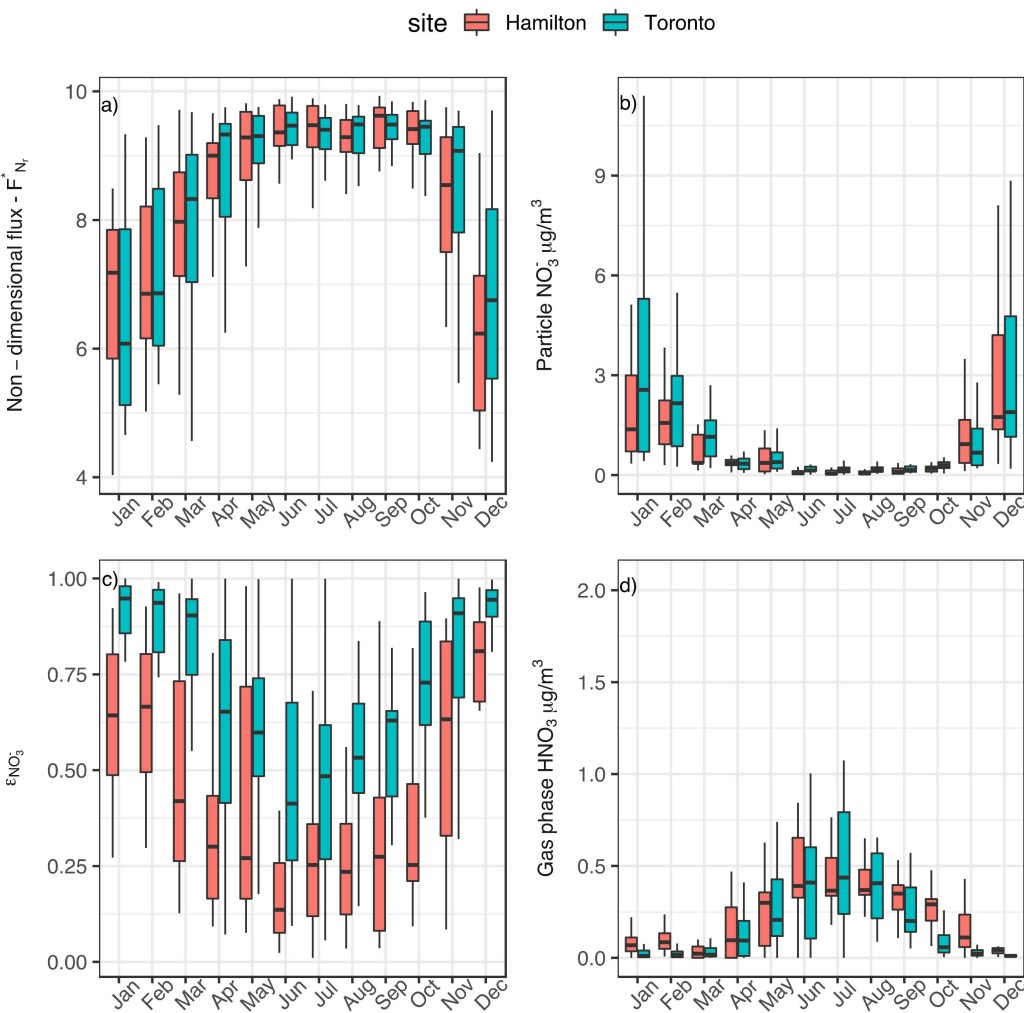

**Figure 6.** Monthly average of (**a**) non-dimensional flux of reactive nitrogen ($N_r$), (**b**) particulate phase $NO_3^-$ concentration in $\mu g\, m^{-3}$; (**c**) gas–particle partitioning coefficient of $NO_3^-$; and (**d**) gas-phase concentration of $HNO_3$ in $\mu g\, m^{-3}$ for Hamilton (red) and Toronto (light blue) sites.

## 4. Conclusions

The results reported in this study indicate the pivotal role played by aerosol pH in regulating the inorganic fraction and mass of aerosol. Indeed, aerosol pH is an important parameter for identifying key aerosol precursors that could be targeted for regional air quality management. For the sites in this work, the inter-seasonal pH variability is mostly driven by temperature changes that cause variations of up to one pH unit. The drop in aerosol acidity from summer to winter indicates a strong sensitivity of aerosol masses to

emissions of $HNO_3$ precursors. Indeed, thermodynamically, these conditions favor $HNO_3$ to be partitioned in the particulate phase.

Water uptake is also critical in determining the aerosol sensitivity to $NH_3$ and $HNO_3$ precursors and deposition regime. In Toronto, particle LWC is dominated by organic species in summer and ammonium nitrate in winter, whereas in Hamilton, the LWC is mostly driven by the inorganic fraction and specifically ammonium nitrate in winter and ammonium sulfate in summer.

The combination of aerosol pH and LWC at the two sites modulates the removal mechanism of N by affecting the N deposition fluxes. While the estimates of $NO_3^-$ deposition fluxes are comparable between the two sites (23.2 µmol m$^{-2}$ s$^{-1}$ in Toronto and 29.2 µmol m$^{-2}$ s$^{-1}$ in Hamilton), due to the higher concentration of NH3 at the Toronto site, the estimates of $NH_3$ deposition fluxes are larger in Toronto (55.5 µmol m$^{-2}$ s$^{-1}$) than in Hamilton (32.5 µmol m$^{-2}$ s$^{-1}$). Variations in LWC and pH also determine the smaller values of the nondimensional N deposition flux ($F_{N_r}^*$) in winter compared to summer. From November to March, N deposition flux slows down leading to an accumulation of N as $NO_3^-$ in the boundary layer. Given the higher aerosol pH in Toronto, aerosol masses in this site are more sensitive to the levels of $HNO_3$ present compared to Hamilton. For both sites, $NO_x$ emissions should be better regulated in order to improve air quality during winter; this is in particular important in Toronto, considering that the aerosol system is thermodynamically more sensitive to the emissions of $HNO_3$ precursors.

**Supplementary Materials:** The following supporting information can be downloaded at: https://www.mdpi.com/article/10.3390/atmos13071012/s1, Figure S1: pH validation plot, predicted versus actual gas-phase ammonia.

**Author Contributions:** Conceptualization, A.M.A. and A.N.; methodology, A.M.A. and A.N.; formal analysis, A.M.A.; visualization, A.M.A.; data curation, P.S. and E.D.-Z.; writing—original draft preparation, A.M.A.; writing—review and editing, A.M.A., A.N., P.S. and E.D.-Z. All authors have read and agreed to the published version of the manuscript.

**Funding:** AA and AN acknowledge the Swiss National Science Foundation project 192292, Atmospheric Acidity Interactions with Dust and its Impacts (AAIDI). AN acknowledges support from project PyroTRACH (ERC-2016-COG) funded from H2020-EU.1.1.—Excellent Science—European Research Council (ERC), project ID 726165.

**Informed Consent Statement:** Informed consent was obtained from all subjects involved in the study.

**Data Availability Statement:** The data presented in this study are openly available in FigShare at 10.6084/m9.figshare.20126087 and 10.6084/m9.figshare.20126120.

**Conflicts of Interest:** The authors declare no conflict of interest.

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
