# Peer review of "Seasonal Aerosol Acidity, Liquid Water Content and Their Impact on Fine Urban Aerosol in SE Canada"

_atmosphere, doi:10.3390/atmos13071012_

Round 1
Reviewer 1 Report
Arangio et al provide a well presented manuscript exhibiting state of the art and with valuable comparison outcomes of experimental data from two Canadian districts with different source apportionment. The experimental data and their presentation, the theoretical background, the validation of the outcomes and the proposition of the current study into the overal particulate matter, pollution regime and health effect perspective is soundly presented. Thus the current manuscript shoudl evaluate some minor corrections for further publication.
Line 45: such sulfur dioxide -> such as sulfur dioxide
Line 81: 1.7 ms -> 1.7 ms-1
Line 188: such road traffic -> such as road traffic
Line 197-198: "However the concentration of particulate SO4- ... character of this pollutant" ->this phrase needs a secondary clarification sentence or change the word "however" in the beginning of the sentence. Hamilton and Toronto are both expected to have SO4- due to smelters and combustion of fossil fuels.
Line 254: Figure numbering and formating needs to be evaluated until the end of the manuscript
Author Response
Arangio et al provide a well presented manuscript exhibiting state of the art and with valuable comparison outcomes of experimental data from two Canadian districts with different source apportionment. The experimental data and their presentation, the theoretical background, the validation of the outcomes and the proposition of the current study into the overal particulate matter, pollution regime and health effect perspective is soundly presented. Thus the current manuscript shoudl evaluate some minor corrections for further publication.
We thank reviewer 1 for the comments. We implemented all reviewer suggestions and address questions as below:
- Line 45: such sulfur dioxide -> such as sulfur dioxide
We thank reviewer 1 for the comment. The typo indicated by reviewer 1 have been updated.
- Line 81: 1.7 ms -> 1.7 ms-1
We thank reviewer 1 for the comment. The typo indicated by reviewer 1 have been updated.
- Line 188: such road traffic -> such as road traffic
We thank reviewer 1 for the comment. The typo indicated by reviewer 1 have been updated.
- Line 197-198: "However the concentration of particulate SO4- ... character of this pollutant" ->this phrase needs a secondary clarification sentence or change the word "however" in the beginning of the sentence. Hamilton and Toronto are both expected to have SO4- due to smelters and combustion of fossil fuels.
We thank reviewer 1 for the comment. We modified the sentence indicating the main result of our analysis, that is, difference of sulfate concertation in PM in the two sites is not statistically relevant being sulfate a regional pollutant. The new sentence is modified as follow:
“The concentrations of particulate SO4- are not statistically different at the two sites, indicating the regional character of this pollutant.”
- Line 254: Figure numbering and formating needs to be evaluated until the end of the manuscript
We thank reviewer 1 to point out the figure numbering and format. We updated and re-checked all figures through the entire manuscript.
Reviewer 2 Report
In this manuscript, Arangio et al reported the aerosol pH and investigated its main drivers and impacts on PM2.5 in two cities in southeast Canada between 2016 and 2018. The study provides some interesting insights. However, the results are not very well organized and presented. Thus, I recommend major revision before considering publication. And I will list the major issues below.
1. What was the seasonal variation of RH at the two sites? I’m not fully convinced by the sensitivity test of pH to temperature when RH could be another important determinant of pH.
2. I find it hard to follow the results and discussion section, the logic is not clear enough with everything in just one section. Suggest separate into subsections. Is the “N Dry Deposition” section part of the results and discussion? The section's title and number are confusing.
3. Line 78-79, missing the figure in the supplementary.
4. Line 268, which is the “purple line”? Do you mean the blue line described in the figure caption?
5. The figures need to be carefully checked. The number of several figures is messed up. For example, between line 254-255, the figure number is wrong, and the x-axis label and title are missing. It seems that the caption is not complete. Line 276, “Figure 3a and 3b” should be Figure 5a and 5b. Between line 341-342, the figure should not be Figure 2. Also, in panel (d), the y-title should not be HNO3-.
6. Line 254, do the authors mean e(NO3-) and e(NH3) instead of the e(NO3) and e(NH4+) which are shown in equation (1)?
7. More explanation of the parameters in each equation is needed. What are Kn1, HHNO3, YH, YNO3- mean in equation (1)? What does the k represent and how does it calculate in equation (2)? Is np the same as the vp described in line 302? If yes, please use the same symbol throughout the context.
8. I don’t understand the equation in line 322, what is F*Nr? And how could F*Nr = F*Nr/npNr?
Author Response
Reviewer 2
In this manuscript, Arangio et al reported the aerosol pH and investigated its main drivers and impacts on PM2.5 in two cities in southeast Canada between 2016 and 2018. The study provides some interesting insights. However, the results are not very well organized and presented. Thus, I recommend major revision before considering publication. And I will list the major issues below.
We thank reviewer 2 for the comments. We implemented the suggestions and answered to the questions from reviewer 2 as below:
- What was the seasonal variation of RH at the two sites? I’m not fully convinced by the sensitivity test of pH to temperature when RH could be another important determinant of pH.
We thank reviewer 2 for this comment. The two sites show a weak or negligible RH seasonality. In addition, in the table below, we report the percentage variation of the monthly RH average (in %) compared to the annual average. RH changes are about 10% which will not be the main contributor for the one order of magnitude variation of [H+].
|
Month |
Hamilton |
Toronto |
|
Jan |
5.1 |
8.3 |
|
Feb |
2.1 |
6.7 |
|
Mar |
2.6 |
-2.6 |
|
Apr |
-7.8 |
-11.8 |
|
May |
1.9 |
-7.4 |
|
Jun |
-9.3 |
-12.5 |
|
Jul |
-6.5 |
-7.9 |
|
Aug |
1.9 |
-2.3 |
|
Sep |
-0.4 |
0.9 |
|
Oct |
2.4 |
9.2 |
|
Nov |
1.3 |
8.9 |
|
Dec |
6.6 |
11.2 |
- I find it hard to follow the results and discussion section, the logic is not clear enough with everything in just one section. Suggest separate into subsections. Is the “N Dry Deposition” section part of the results and discussion? The section's title and number are confusing.
We thank reviewer 2 for this comment to improve the readability of the manuscript. We renumbered the section and split the result and discussion into two subsections: “3.1 Seasonality of aerosol acidity” and “3.2 Implication of aerosol acidity variability on Nitrogen Dry Deposition.
- Line 78-79, missing the figure in the supplementary.
We add the supplementary material separately.
- Line 268, which is the “purple line”? Do you mean the blue line described in the figure caption?
The indication from reviewer 2 has been updated. For “purple line” we mistakenly referred to the “red” diagonal line that cross the plot in figure 4 as described in the caption.
- The figures need to be carefully checked. The number of several figures is messed up. For example, between line 254-255, the figure number is wrong, and the x-axis label and title are missing. It seems that the caption is not complete. Line 276, “Figure 3a and 3b” should be Figure 5a and 5b. Between line 341-342, the figure should not be Figure 2. Also, in panel (d), the y-title should not be HNO3-.
The indications from reviewer 2 have been updated. All figures have been carefully re-checked and corrected.
- Line 254, do the authors mean e(NO3-) and e(NH3) instead of the e(NO3) and e(NH4+) which are shown in equation (1)?
We thank the reviewer 2 for pointing out this typo. We updated the manuscript indicating the two parameters indicated by reviewer 2 as and to indicate the gas-particle partitioning of the pair of species HNO3/NO3- and NH3/NH4+ respectively.
- More explanation of the parameters in each equation is needed. What are Kn1, HHNO3, YH, YNO3- mean in equation (1)? What does the k represent and how does it calculate in equation (2)? Is npthe same as the vp described in line 302? If yes, please use the same symbol throughout the context.
We thank the reviewer 2 for pointing out the missing parameter description. We updated the manuscript as follow:
“where and and the Henry’s law constants for HNO3 and NH3, respectively; Kn1 and Ka are the acid dissociation constants for HNO3 and NH4+; R is the universal gas constant; and , and are the single-ion activity coefficients for H+ and NO3-, respectively.”
The np in equation 3 was the same as vp as indicated in the main text. We included the description of equation 3 as follow:
“Where Nr is the total reactive nitrogen concentration, is the fraction of Nr in the form of .
- I don’t understand the equation in line 322, what is F*Nr? And how could F*Nr= F*Nr/npNr?
We thank reviewer 2 to point out this typo. We updated the equation and the description as follow:
“In addition, considering as the sum of FNO3T and FNH3T, the non-dimensional flux for reactive nitrogen (Nr) can be derived as follow:
)
Where is the non-dimensional version of the reactive nitrogen flux as indicated in the description before.

Round 2
Reviewer 2 Report
Agree to publish on the current form.